# Automated Segmentation of Left Ventricle in 2D echocardiography using deep learning

**Neda Azarmehr**[1,2]                                               nAzarmehr@lincoln.ac.uk
**Xujiong Ye**[1]                                                           xye@lincoln.ac.uk
**Faraz Janan**[1]                                                      fjanan@lincoln.ac.uk
**James P Howard**[2]                                      james.howard1@imperial.ac.uk
**Darrel P Francis**[2]                                          d.francis@imperial.ac.uk
**Massoud Zolgharni**[2,3]                                 massoud.zolgharni@uwl.ac.uk

[1] *School of Computer Science, University of Lincoln, Lincoln, UK*

[2] *National Heart and Lung Institute, Imperial College London, London, UK*

[3]*School of Computing and Engineering, University of West London, London, UK*

## Abstract

Following the successful application of the U-Net to medical images, there have been different encoder-decoder models proposed as an improvement to the original U-Net for segmenting echocardiographic images. This study aims to examine the performance of the state-of-the-art proposed models claimed to have better accuracies, as well as the original U-Net model by applying them to an independent dataset of patients to segment the endocardium of the Left Ventricle in 2D automatically. The prediction outputs of the models are used to evaluate the performance of the models by comparing the automated results against the expert annotations (gold standard). Our results reveal that the original U-Net model outperforms other models by achieving an average Dice coefficient of 0.92±0.05, and Hausdorff distance of 3.97±0.82.

**Keywords:** Echocardiography, Segmentation, Deep Learning

## 1. Introduction

To assess the cardiac function in 2D ultrasound images, quantification of the Left Ventricle (LV) shape and deformation are crucial, and this relies on the accurate segmentation of the LV contour in end-diastolic (ED) and end-systolic frames (Raynaud et al., 2017). At present, the manual segmentation of the LV suffers from various complications: (i) it needs to be carried out only by an experienced clinician; (ii) inevitable inter- and intra-observer variability in the annotations; (iii) and it is laborious and must be repeated for each patient. Consequently, the automatic segmentation methods have been designed to resolve this issue that can lead to increased patient throughput and can reduce the inter-user discrepancy. There are many suggested methods for 2D LV segmentation. Recently Deep Convolutional Neural Networks have shown promising results for image segmentation (Jafari et al., 2018; Leclerc et al., 2019), specifically U-Net, which has been successfully applied to multiple medical image segmentation problems. As explained in the abstract, this study aims to investigate and compare the performance of U-Net 1 and U-Net 2 models reported by

(Leclerc et al., 2019) with the original U-Net (Ronneberger et al., 2015). More details in Appendix A.

## 2. Methods and dataset

The study population consisted of 61 patients with Apical 4-chamber views, who were recruited from patients who had undergone echocardiography with Imperial College Healthcare NHS Trust. The study was approved by the local ethics committee and written informed consent was obtained. All data acquired with the same equipment. To achieve the gold-standard (ground-truth) measurements, one accredited and experienced cardiology expert manually traced the LV borders. Out of 1098 available frames (61 patients ×3 positions ×3 heartbeats ×2 ED/ES frames), a total of 992 frames were annotated. To investigate the inter-observer variability, a second operator repeated the LV tracing on 992 frames, blinded to the judgment of the first operator. From the total of 992 images, 60% were selected for training, 20% of total data used for validation, and the remaining 20% was used for testing. All three datasets comprised images from different patients and no images from the same patient were shared between the datasets.

All images were resized to a smaller dimension of $256 \times 256$ pixels (as needed by model U-Net 1 and U-Net 2), leading to a fair comparison. All models produce the output with the same spatial size as the input image. Pytorch was used for the implementations (Paszke et al., 2017), where Adam optimiser with 250 epochs and learning rate of 0.00001 were used for training the models. The negative log-likelihood loss is used as the networks objective function. All computations were carried using an Nvidia GeForce GTX 1080 Ti GPU.

## 3. Experiment results and discussion

All models were trained separately using the annotations provided by either of the operators. The Dice Coefficient (DC) and Hausdorff distance (HD) were employed to evaluate the performance and accuracy of the models in segmenting the LV region. Fig 1A displays output examples from the three models when trained using annotation provided by Operator-A (i.e., OA). To specify the LV endocardium border, the contour of the predicted segmentation was used. The solid blue line indicates the manual annotation while the yellow line shows the automated results. As can be seen, the U-Net model achieved a higher DC (0.97) and lower HD (4.24) score. A visual inspection of the automatically detected LV border also confirms this. The LV border obtained from the U-Net 1 and U-Net 2 models seems to be less smooth compared to that in the U-Net model. However, all three models look to perform with reasonable accuracy.

Fig 1B illustrates the results for a failed case example, for which all 3 models seem to struggle. It is evident that image quality tends to be lower in failed cases due to the missing borders, the presence of speckle noise or artefacts, and poor contrast between the myocardium and blood pool. The left side of Table 1, provides average DC and HD for the 3 models, across all 199 testing images. Plausible scenarios for manual or automated (U-Net only) are provided on the right side of the table; for each image, there were 4 assessments of the LV border; 2 human and 2 automated (trained by annotation of either of human operators). The automated model performs similarly to human operators. The model

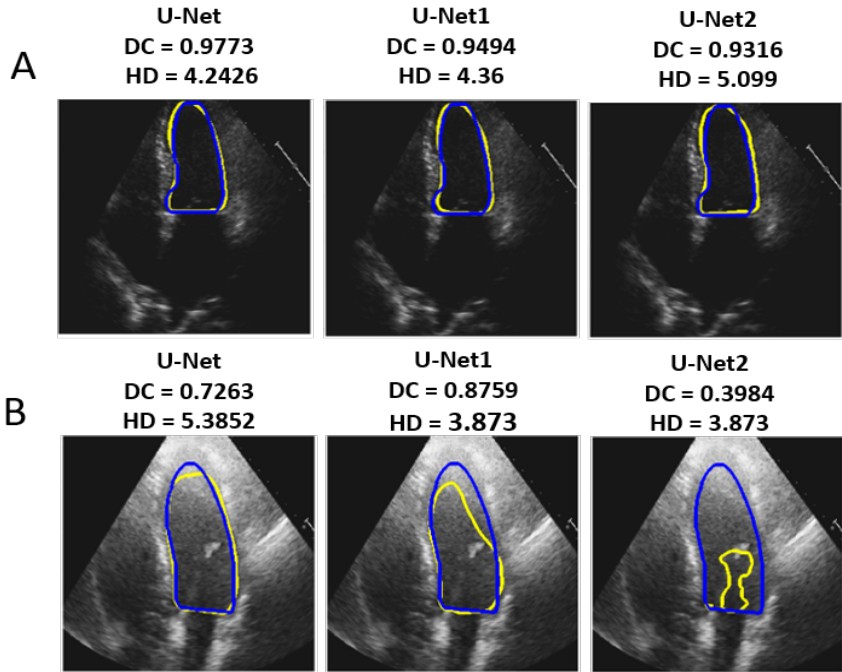

Figure 1: Good and failed case example outputs from three different models.

disagrees with the Operator-A, but so does the Operator-B. Since different experts make different judgments, it is not possible for any automated model to agree with all experts. However, it is desirable for the automated models not to have larger discrepancies when compared with the performance of human judgments; that is, to behave approximately as well as human operators.

Table 1: Comparison of evaluation measures expressed as mean±SD for all 3 models, and 5 possible scenarios for U-Net only. OA is the Operator-A, and POA is the predicted results by a model trained by gold-standard from Operator-A.

| Model | DC | HD | Compared Scenarios | DC | HD |
|---|---|---|---|---|---|
| U-Net | $0.92 \pm 0.05$ | $3.97 \pm 0.82$ | OA VS OB | $0.88 \pm 0.06$ | $4.50 \pm 0.87$ |
| | | | POA VS OA | $0.92 \pm 0.05$ | $3.97 \pm 0.82$ |
| U-Net 1 | $0.92 \pm 0.04$ | $4.16 \pm 0.73$ | POA VS OB | $0.90 \pm 0.05$ | $4.08 \pm 0.91$ |
| | | | POB VS OB | $0.91 \pm 0.06$ | $4.24 \pm 0.75$ |
| U-Net 2 | $0.90 \pm 0.12$ | $4.09 \pm 0.80$ | POB VS OA | $0.89 \pm 0.07$ | $4.14 \pm 0.80$ |

## 4. Conclusion

Our study compared the performance of three models which appear to perform no worse than human experts. However, the automated models, when encountered with the lower image qualities, demonstrate larger discrepancies with the gold-standard annotations. This is potentially caused by the lack of balanced data in terms of different levels of image qualities in our dataset. Future investigations will examine the correlation between the performance of the automated model and the image qualities.

## Acknowledgments

N.A. was supported by the School of Computer Science PhD scholarship at the University of Lincoln.

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

## Appendix A.

Details of models summaries in the following table. The number of feature maps given in table corresponds to the number of convolutions in the convolution layers. For each U-Net implementation, the values for the first, the bottom (where the spatial information is the most compressed), and the last convolution layers indicated (Leclerc et al., 2019).

| Architectures | Number of feature maps | Upsampling scheme | Normalization scheme |
|---|---|---|---|
| U-Net | $64 \downarrow 1024 \uparrow 64$ | Deconvolution | None |
| U-Net 1 | $32 \downarrow 128 \uparrow 16$ | 2 ×2 repeats | None |
| U-Net 2 | $48 \downarrow 768 \uparrow 48$ | Deconvolutions | BatchNorm |

