# OpenReview forum: "Automated Segmentation of Left Ventricle in 2D echocardiography using deep learning"
_MIDL.io/2019/Conference/Abstract — MIDL Abstract 2019_

### Official Review · AnonReviewer1 · 2019-04-24
**Great dataset**

**Rating:** 2
**Confidence:** 3

**Review:**

The author has made a great effort to prepare the training data for segmenting left ventricle.
The major limitation of the work is the unclearness and novelty of the method.
First, the definition of U-Net 1 and 2 are not quite clear from the paper.
It seems the novelty on improving U-Net is limited.

---

### Official Review · AnonReviewer2 · 2019-04-30
**Comparison of 3 U-net architectures for segmentation of the left ventricle in 2D echocardiography**

**Rating:** 3
**Confidence:** 3

**Review:**

From a set of 62 patients 3 slices were segmented by two independent observers.
The networks were trained with either one of the segmentations and results were compared, also with manual annotations by the observers.
The paper is clearly written, the method and experiments seem sound, reported performance is good.

---

### Decision · Program_Chairs · 2019-05-06
**Acceptance Decision**

Accept